# Healthcare resource utilization and associated costs among patients with migraine in Finland: A retrospective register-based study

**Mikko Kosunen**[1], **Jari Rossi**[1], **Severi Niskanen**[2], **Roope Metsä**[1], **Ville Kainu**[1], **Mari Lahelma**[2], **Outi Isomeri**[2]*

**1** Pfizer Oy, Helsinki, Finland, **2** NHG Finland, Nordic Healthcare Group, Helsinki, Finland

* outi.isomeri@nhg.fi

**Data Availability Statement:** All relevant data are within the manuscript and its Supporting information files.

## Abstract

Migraine is a common chronic brain disorder, characterized by recurring and often disabling attacks of severe headache, with additional symptoms such as photophobia, phonophobia and nausea. Migraine affects especially the working age population. The objective of this retrospective observational register-based study was to analyze the use of healthcare services and associated costs in Finnish migraine patients. Study was based on aggregate data from January 1st, 2020, to December 31st, 2021, from the Finnish Institute for Health and Welfare's national registries. Patients were grouped into nine patient groups according to medication prescriptions and diagnoses. Healthcare resource utilization in specialty, primary, and occupational healthcare was assessed and analyzed separately for all-cause and migraine related healthcare contacts from a one-year period. The total number of patients was 175 711, and most (45%) of the patients belonged to a group that had used only one triptan. Migraine related total healthcare resource utilization was greater for patients that had used two or more triptans compared to those that had used only one. The patients with three or more preventive medications had the highest total migraine related healthcare resource utilization of the studied patient cohorts. Of the total annual healthcare costs 11.5% (50.6 million €) was associated to be migraine related costs. Total per patient per year healthcare costs were highest with patients that had used three or more preventive medications (5 626 €) and lowest in those with only one triptan (2 257 €). Our findings are in line with the recent European Headache Federation consensus statement regarding the unmet need in patients who have had inadequate response to two or more triptans. When assessing the patient access and cost-effectiveness of novel treatments for the treatment of migraine within different healthcare systems, a holistic analysis of the current disease burden along with potential gains for patients and healthcare service providers are essential information in guiding decision-making.

## Introduction

Migraine is a common chronic brain disorder characterized by recurring and often disabling attacks of severe headache, accompanied with additional symptoms such as aura, sensitivity to

**Funding:** The study was sponsored by Pfizer Oy. MK, JR, RM and VK are employed by Pfizer Oy and own Pfizer stocks. OI, SN and ML are employees of Nordic Healthcare Group, which received funding from Pfizer Oy in connection with the development of this manuscript. Role of Funder statement: The funders participated in study design, analysis, decision to publish and preparation of the manuscript. The funders had no role in the data collection.

**Competing interests:** There are no competing interests to disclose.

light and sound, and gastrointestinal symptoms [1, 2]. Migraine is more prevalent in women than in men, with roughly 19% of women and 9% of men worldwide suffering from migraine [3]. This condition particularly affects the working-age population, and overall, it is most observed between the ages of 20 and 64 [4].

Migraine has a significant negative impact on daily activities, healthcare resource utilization (HCRU), and work ability. An analysis of Finnish respondents to the My Migraine Voice survey noted an average of 5.9 doctor's visits per month over the preceding six months. Furthermore, a vast majority of the respondents reported that migraine had a negative impact on their working life, with nearly half reporting a decrease in work productivity [5].

The choice of treatment for migraine attacks according to current guidelines is influenced by the severity of the attacks, presence of nausea, vomiting, time of onset, coexisting conditions, other illnesses, and the expected side effects of the treatment. According to Finnish Current Care Guidelines, mild or moderate migraine attacks can be treated with paracetamol or NSAIDs, either alone or in combination with metoclopramide. For more severe attacks, triptans should be taken immediately, rather than waiting for first-line NSAIDs to prove ineffective [6]. All triptans are serotonin 1B/1D receptor agonists, inhibiting the release of several neuropeptides including calcitonin gene-related peptide (CGRP) from the sensory nerve endings of the trigeminal nerve, and constricting the dilated cerebral blood vessels in migraine attacks [1].

Although triptans are considered suitable for most patients with migraine, meta-analysis-based estimates indicate that only approximately 18–33% achieve pain relief for 24 hours with triptans, and 20–34% require additional medication for relapse [7]. It is also worth noting that switching to another triptan is not necessarily associated with improvements in headache-related disability and may only lead to increased healthcare costs [8, 9].

According to the European Headache Federation consensus statement on the treatment of migraine headache, an effective treatment is considered to be 1) improvement from severe to moderate to mild or resolution of headache, 2) no or minimal symptoms related to migraine but unrelated to pain, and 3) no significant medication-related side effects within two hours of taking the medication and maintaining this status for at least 24 hours [10]. For triptans, a response is considered when the drug proves effective in three out of four consecutive attacks. On the other hand, an incomplete response is proposed to be divided into three categories: 1) incomplete response to one triptan (triptan non-responder), 2) incomplete response to two triptans (triptan-resistant), and 3) incomplete response to ≥3 triptans, one of which is a subcutaneous preparation (triptan-refractory). Among these, patients with inadequate response to two or more triptans (triptan-resistant and triptan-refractory) are considered to have a particular medical need for new migraine medications such as gepants [10].

Preventive medication is used to reduce the frequency, severity, and duration of migraine attacks, and include drugs such as beta and potassium blockers, tricyclic antidepressants, anti-epileptics, and drugs affecting the renin-angiotensin system [2]. The importance of CGRP signaling in the neurobiology of migraine and in the development of monoclonal antibodies (mABS) based on inhibition of CGRP, or its receptor activation for migraine profylaxis has become central due to its specificity and potentially lower risk of adverse events [11]. According to Finnish Current Care Guidelines, the first line drugs for prophylactic treatment of migraine are non-ISA betablockers or candesartan and amitriptyline [6]. Unfortunately, most chronic migraine patients taking oral prophylactics discontinue medication within 6 months of starting the medication, and more than 60% of patients have a history of one or more failed attempts [12, 13]. Previous studies have also suggested that patients with a history of failed preventive treatments for migraine have higher utilization of healthcare resources [14].

In Finland every citizen is entitled to publicly funded specialty and primary healthcare services. An estimated 2 million inhabitants are also covered by occupational healthcare (OHC).

OHC is financed by customer fees paid by the employer, and it therefore differs from the financing of other publicly funded social and healthcare services [15, 16].

The objective of this study was to analyze the use of healthcare services and associated costs in Finnish migraine patients based on their acute- and preventive migraine medication and diagnosis history and elucidate demographics and differences between different patient groups.

## Material and methods

### Study design

This retrospective register-based study was based on aggregate data from the Finnish Institute for Health and Welfare's (THL) registers Care Register for Health Care (Hilmo) and Register of Primary Health Care (Avohilmo). These two registers are national care notification registers of Finland collecting data from all institutions providing specialized and inpatient care (Hilmo) and primary outpatient care (Avohilmo). In addition to public healthcare, registers also cover service use in the private and occupational sector, where significant amount of migraine patients are treated in Finland [16]. The study population was identified from Hilmo and Avohilmo registers between January 1$^{st}$ 2020 to December 31$^{st}$ 2021 and patient groups were formed as illustrated in Fig 1. The summary level data for HCRU of the patient groups were obtained from one-year period between January 1$^{st}$ 2021 to December 31$^{st}$ 2021.

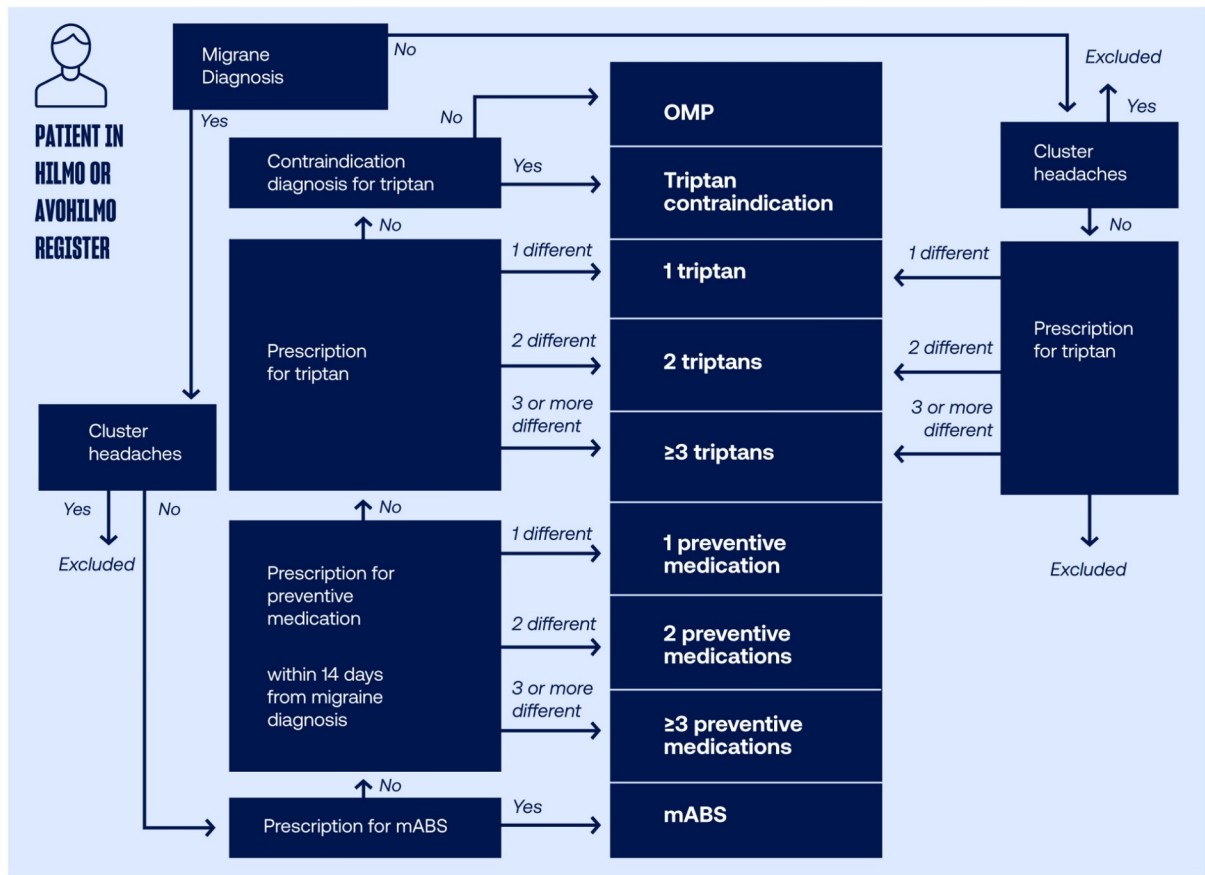

**Fig 1. Patient flowchart for the identification of patient groups.** mABS, monoclonal antibodies; OMP, Other Migraine Patients.

The data from Hilmo and Avohilmo registers were combined by the registry holder on a patient level through unique patient identification codes and aggregated on a patient group level to assess HCRU of patients with migraine.

## Study population

The inclusion period for study population from Hilmo and Avohilmo registers was from January 1st 2010 to December 31st 2021. The patients with migraine were identified based on following criteria a) diagnosis of migraine (based on ICD-10 code G43 or ICPC-2 code N87 used in primary healthcare), or prescribed triptan medication (based on ATC-codes N02CC*). Patients with cluster headaches (based on ICD-10 code G44.0 or ICPC-2 code N90) were excluded, since these patients may also be treated with triptan medication in Finland.

Patients included in the study were grouped into nine different patient groups according to medication prescriptions and diagnoses as illustrated in Fig 1. First, each patient with migraine diagnosis was hierarchically included in one of the following patient groups based on the first met criteria: (1) prescription for CGRP mABS (mABS group), (2) prescription for at least three different preventive medications with identified migraine diagnosis no more than two weeks prior each prescription (≥3 preventive medications group), (3) prescription for two different preventive medications with identified migraine diagnosis no more than two weeks prior both prescriptions (2 preventive medications group), (4) prescription for one preventive medication with identified migraine diagnosis no more than two weeks prior prescription (1 preventive medication group), (5) prescription for at least three different triptans (≥3 triptans group), (6) prescription for two different triptans (2 triptans group), (7) prescription for one triptans (1 triptan group), (8) contraindication diagnosis for triptans (triptan contraindication group), or (9) none of the previous but as a prerequisite, the migraine diagnosis within 2 years (Other Migraine Patients [OMP] group) (Fig 1). Second, patients without migraine diagnosis but with prescribed triptan were hierarchically included in one of the following patient groups based on the first met criteria: (1) prescription for at least three different triptans (≥3 triptans group), (2) prescription for two different triptans (2 triptans group), or (3) prescription for one triptan (1 triptan group) (Fig 1). The coding classifications used for diagnoses (migraine, cluster headaches, triptan contraindications) and medications (mABS, preventive medications, triptans) are listed in Supplemental materials (S1 Table). The list of contraindications analyzed in this study was adapted from the summary of product characteristics of triptans available in Finland.

## Healthcare resource utilization

Healthcare utilization was assessed in specialty healthcare (SHC), primary healthcare (PHC), and occupational healthcare (OHC). Outpatient visits (other than emergency visits), emergency visits and inpatient stays were assessed for SHC and PHC, and total number of outpatient visits for OHC. SHC and PHC visits were identified from separate registers and included only physical visits as telephone and digital contacts could not be identified. Emergency visits were identified by visits with a notification of "emergency" as a specialty including also visits defined as urgent. SHC and PHC inpatient stays, which included hospital admissions, were identified based on entry and exit dates. OHC visits were identified based on OHC-specific service type codes. All PHC components and OHC were identified from Avohilmo register and SHC from Hilmo register. All-cause and migraine related HCRU (based on ICD-10 code G43 or ICPC-2 code N89) were assessed separately. Both total annual and per patient HCRU were reported.

## Associated costs

HCRU and associated costs of outpatient and inpatient contacts in SHC and PHC during 2021 for all studied patient cohorts were estimated. PHC visits also include visits to OHC and these were estimated separately. All unit costs used in calculations correspond to a physical visit. HCRU associated unit costs in PHC and SHC were assessed separately for migraine associated visits and non-migraine associated, other visits. For migraine associated outpatient visits, emergency visits, and inpatient days in SHC the average of all hospital's (including university hospitals, central hospitals, and healthcare centers managed by specialists) neurology specific unit costs were used. A headache specific unit cost was used for migraine associated outpatient visits in PHC. Outpatient visits, emergency visits, and inpatient days in SHC for non-migraine contacts were valued with the average unit cost of all SHC hospitals. PHC outpatient visit was valued as an average cost of a physician visit. All outpatient emergency visits in PHC were associated with an additional emergency on-call fee. No migraine specific unit cost for PHC inpatient days or OHC visits was available. All PHC inpatient days were valued with a daily cost derived from short-term care episodes (<90 days/episode). The average unit cost of an OHC visit was valued as the average of OHC physician visit and OHC specialist visit since these visits could not be separated from the data. All unit costs were sourced from national health and social care unit cost report and inflated to 2021 value using healthcare price index from Statistics Finland [17, 18]. Total HCRU and associated costs of patients with migraine were calculated by multiplying each healthcare contact by the associated unit cost. Costs were calculated as total per year and total per patient per year.

All analyses were performed using Microsoft Excel 2020 (Redmond, WA, USA). All analyses are based on aggregate data, and the results are reported as mean (SD) or number (%), as stated. Given the descriptive nature of the study, no inferential statistics were performed.

## Results

### Study population

Demographic characteristics of the patients are shown in Table 1. The total number of patients was 175 711, and most of them were included into the 1 triptan group (n = 78 327, 45%) or the OMP group (n = 56 090, 32%) (Table 1). Majority of the patients in all groups

**Table 1. Demographic characteristics of the study population.**

|  | OMP | Triptan contr | Triptan | 2 triptans | ≥3 triptans | Prev | 2 prev | ≥3 prev | mABS |
|---|---|---|---|---|---|---|---|---|---|
| **Number of patients** | 56 090 | 5 287 | 78 327 | 7 186 | 1 064 | 18 367 | 3 262 | 591 | 5 537 |
| **Sex, %** |  |  |  |  |  |  |  |  |  |
| **Women** | 74.8% | 72.7% | 78.7% | 84.2% | 85.7% | 82.1% | 85.4% | 86.7% | 87.4% |
| **Age, years, *mean*** | 33.7 | 59.8 | 40.9 | 39.4 | 38.5 | 38.3 | 39.3 | 38.8 | 43.9 |
| **Age, years, *median*** | 31.0 | 61.4 | 40.5 | 39.1 | 38.2 | 37.4 | 38.6 | 37.1 | 44.0 |
| **Age, years, %** |  |  |  |  |  |  |  |  |  |
| **0–17** | 17.7% | 0.9% | 5.9% | 4.5% | 4.1% | 9.8% | 7.2% | 6.6% | 1.3% |
| **18–34** | 40.5% | 8.6% | 31.5% | 35.0% | 36.9% | 34.9% | 35.1% | 38.2% | 23.9% |
| **35–49** | 24.1% | 18.3% | 34.1% | 38.0% | 39.5% | 32.2% | 33.2% | 32.3% | 42.3% |
| **50–64** | 12.7% | 29.1% | 22.7% | 19.6% | 17.4% | 17.3% | 17.9% | 17.1% | 27.8% |
| **65+** | 4.9% | 43.1% | 5.8% | 3.1% | 2.2% | 5.9% | 6.7% | 5.8% | 4.6% |

contr, contraindication; mABS, monoclonal antibodies; OMP, Other Migraine Patients.

prev, preventive medication.

were women. The mean age was highest in the triptan contraindication group and lowest in the OMP group. Especially the percentage of patients <18 years old in the OMP group was notably high.

## Healthcare resource utilization

The overview of the migraine related and all-cause HCRU per patient group is presented in Fig 2. Migraine related total HCRU was greater for patients that had used two or more triptans compared to those that had used only one (Fig 2). The patients that had used three or more preventive medications had the highest total migraine related HCRU of the studied patient cohorts (Fig 2).

The migraine related HCRU components per patient group are presented in Fig 3. Migraine related SHC and OHC visits were greater for patients that had used two or more triptans compared to those that had used only one. The patients that had used three or more preventive medications had the highest migraine related service usage in all HCRU components of the studied patient cohorts.

The all-cause HCRU components of the patient groups are presented in Fig 4. The patients that had used three or more preventive medications had the highest number of all-cause SHC and PHC visits while the number of OHC visits for any cause was highest among patients who had used mABS.The differences in all-cause HCRU between patients with one triptan and those with two or at least three triptans were minor.

The proportion of migraine related total HCRU of all-cause total HCRU was 8.5% in the OMP group, 9.1% in triptan contraindication group, 4.0% in 1 triptan group, 7.3% in 2 triptans group, 11.6% in ≥3 triptans group, 14.5% in 1 preventive medication group, 21.0% in 2 preventive medications group, 26.5% in ≥3 preventive medications group and 24.9% in mABS group.

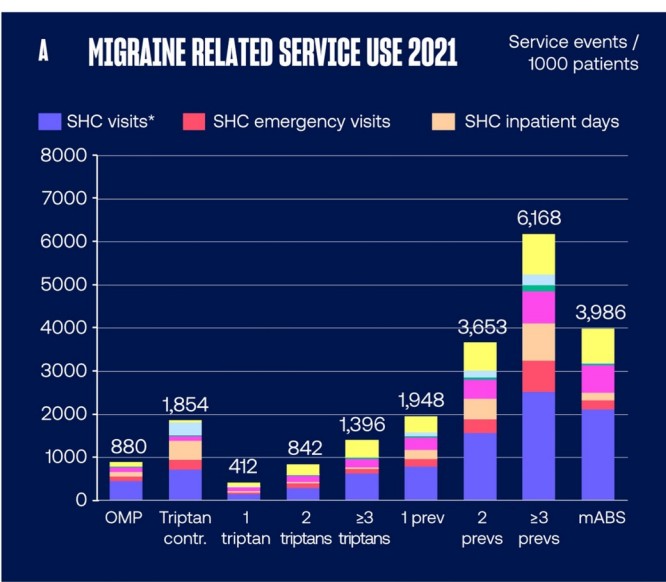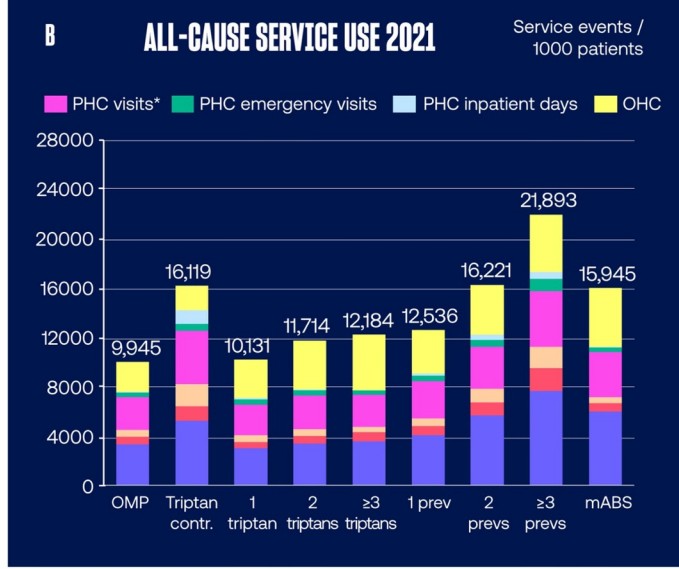

**Fig 2.** Annual (A) migraine related and (B) all-cause healthcare resource utilization. Data are shown as number of service events per 1000 patients. *Other than emergency visits. contr, contraindication; mABS, monoclonal antibodies; OHC, occupational healthcare; OMP, Other Migraine Patient; PHC, primary healthcare; prev, preventive medication; SHC, specialty healthcare.

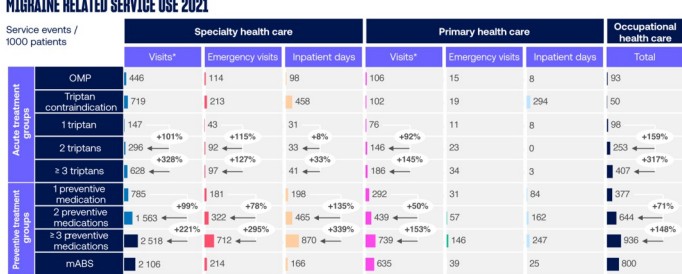

**Fig 3. Annual migraine related healthcare resource utilization (HCRU) by HCRU component.** HCRU of patients with 2 or ≥3 triptans were compared to patients with 1 triptan, and HCRU of patients with 2 or ≥3 preventive medications were compared to patients with 1 preventive medication. Data are shown as number of service events per 1000 patients. *Other than emergency visits. mABS, monoclonal antibodies; OMP, Other Migraine Patient.

## Associated costs

Total annual all-cause healthcare costs and total annual migraine related heathcare costs are shown in Table 2. Annual total all-cause healthcare costs for patients with migraine were estimated to be nearly 439.8 million €. Of the total healthcare costs 11.5% (50.6 million €) was associated to be migraine related costs. Patients in OMP group and in triptan contraindication group were associated with a greater migraine share of total costs compared to patients who used 1, 2 or ≥3 triptans. Patients with ≥3 preventive medications were associated with a greater migraine share of total costs compared to patients that had used 1 or 2 preventive medications or had used mABS -medication. Total annual all-cause healthcare costs and total annual migraine related healthcare costs stratified by HCRU components and patient groups are shown in Supplemental material (S2 and S3 Tables, respectively).

Annual healthcare costs per patient stratified to migraine related and other costs are shown in Fig 5. Total per patient per year healthcare costs were highest with patients that had used three or more preventive medications (5 626 €) or had triptan contraindications (4 556 €), and lowest in those with prescription to only one triptan (2 257 €). The highest migraine related per patient per year costs (1 904 €) and percentage of migraine related costs of total costs (33.8%) were also seen for patients with ≥ 3 preventive medications.

Migraine related per patient per year costs were 6.4-fold higher for patients with triptan contraindication,1.9-fold higher for patients with 2 triptans and 3.3-fold higher for patients

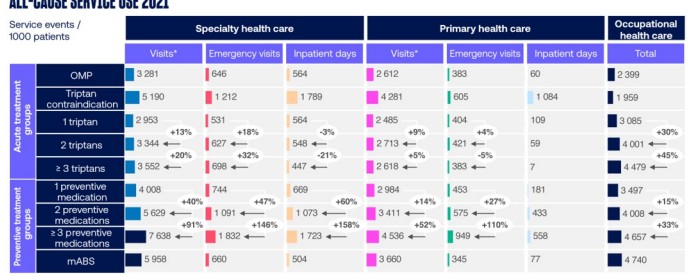

**Fig 4. Annual all-cause healthcare resource utilization by healthcare resource utilization component.** HCRU of patients with 2 or ≥3 triptans were compared to patients with 1 triptan, and HCRU of patients with 2 or ≥3 preventive medications were compared to patients with 1 preventive medication. Data are shown as number of service events per 1000 patients. *Other than emergency visits. mABS, monoclonal antibodies; OMP, Other Migraine Patient.

**Table 2. Annual total all-cause and migraine related healthcare costs (rounded to nearest 1000€).**

| Patient group | Patients (n) | All-cause healthcare costs | Migraine related healthcare costs | Migraine share (%) of total costs |
|---|---|---|---|---|
| **OMP** | 56 090 | 129 917 000 | 15 875 000 | 12.2% |
| **Triptan contraindication** | 5 287 | 24 089 000 | 3 651 000 | 15.2% |
| **1 triptan** | 78 327 | 176 770 000 | 8 432 000 | 4.8% |
| **2 triptans** | 7 186 | 18 083 000 | 1 492 000 | 8.3% |
| **≥3 triptans** | 1 064 | 2 708 000 | 373 000 | 13.8% |
| **1 preventive medication** | 18 367 | 52 452 000 | 10 170 000 | 19.4% |
| **2 preventive medications** | 3 262 | 12 986 000 | 3 550 000 | 27.3% |
| **≥3 preventive medications** | 591 | 3 325 000 | 1 125 000 | 33.8% |
| **mABS** | 5 537 | 19 457 000 | 5 907 000 | 30.4% |
| **Total** | **175 711** | **439 787 000** | **50 575 000** | **11.5%** |

mABS, monoclonal antibodies; OMP, Other Migraine Patient.

with ≥ 3 triptans compared to patients who had used only 1 triptan, respectively. In the preventive medication populations, migraine related per patient per year costs were 2.0-fold higher for patients with 2 preventive medications, 3.4-fold higher for patients with ≥ 3 preventive medications and 1.9-fold higher for patients with mABS compared to patients who had used only 1 preventive medication, respectively.

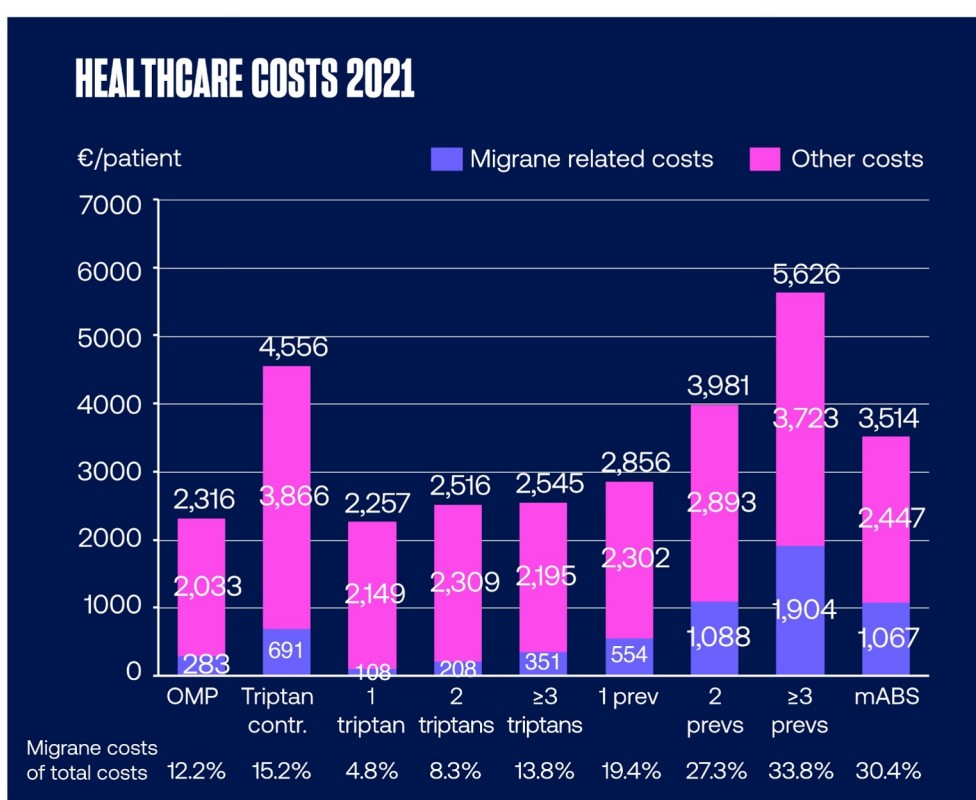

**Fig 5. Annual healthcare costs per patient.** Data are shown as mean € per patient per year and stratified into migraine related and other costs. contr, contraindication; mABS, monoclonal antibodies; OMP, Other Migraine Patient.

The breakdown of migraine related healthcare contacts and costs into different HCRU components are shown in Supplemental material (S4 and S5 Tables, respectively). SHC visits (excluding ER visits) accounted for 43.9% of total migraine related healthcare contacts and 52.5% of total migraine related costs. SHC ER visits accounted for 10.2% and 19.7%, SHC inpatient days for 10.0% and 14.2%, PHC visits (excluding ER visits) for 14.2% and 3.1%, PHC ER visits for 1.7% and 0.6%, PHC inpatients days for 2.9% and 3.8% and occupational healthcare for 17.0% and 6.1% of the total migraine related healthcare contacts and costs, respectively.

Occupational healthcare accounted for the largest share of total migraine related healthcare contacts and costs in the 2 triptans (30.0% and 12.8%) and ≥3 triptans (29.2% and 12.2%) groups. In the mABS group, SHC visits accounted for the largest share (52.8% and 69.3%) and SHC emergency room visits for the smallest share (5.4% and 11.4%) of total migraine related contacts and costs, respectively.

Of the migraine related contacts and costs, the share of SHC inpatient days and costs (24.7% and 27.7%) and PHC inpatient days and costs (15.8% and 16.4%) was largest, and the share of occupational healthcare contacts and costs (2.7% and 0.8%) smallest in the triptan contraindication group. The average age (59.8 years) of this patient cohort was almost 20 years higher compared to other patient groups. The share of over 65-year-old patients (43.1%) was also high, indicating that patients in the triptan contraindication cohort include a significant amount of patients outside of occupational healthcare services compared to other groups.

## Discussion

To our knowledge, this is the first study from national registries in Finland showing the distribution of migraine patients based on medication usage and the related healthcare resource utilization in different patient groups. Previously, the burden of migraine in Finland has been studied in occupational healthcare and this study similarly showed that the disease burden increased by each failed treatment line and severity of migraine [5, 16]. In this study, we have used the most recent data accrued from nationwide Hilmo and Avohilmo registers, which contain data from all scheduled and emergency care inpatient and outpatient care visits, and primary healthcare contacts at health centers. This study strengthens the previous observations by elucidating the total healthcare utilization and associated costs among patients with migraine in Finland.

Estimating the burden of migraine based on HCRU and associated costs varies considerably between different healthcare systems in different countries and depending on different study methodologies used and data resources and data coverage available at different timepoints. In a recent longitudinal, multicohort, web-based survey in the United States it was found that the most common locations for migraine care were primary care, neurologist, and emergency department settings, with consultation in the primary care being the most common type of consultation [19]. In our study with national coverage, migraine care was found to be more focused on the specialty care setting. The total migraine related SHC contacts from all migraine related contacts accounted from 49.9% for patients in the 2 triptans group up to 66.5% and 75.0% in the ≥3 preventive medications and triptan contraindication groups, respectively. The reason for these differences may be explained by differences in study type and data coverage, but also explained by different healthcare systems and access to healthcare specialists between Finland and the United States.

Recent studies from Finland, utilizing data from same THL registries as our study, have found that the total annual direct HCRU costs for individuals with obesity were 1 980 € per person [20] and that the mean direct hospital and outpatient costs 1–5 years after treatment-

resistant depression were 6 357–2 333 € per year in patients with treatment-resistant depression and 2 966–1 658 € per year in patients with non-treatment-resistant depression [21]. However, although recent, these studies include time periods when occupational healthcare data was not yet included in THL registries. Therefore, also the comparison of per patient costs for different diseases from the same registries, but covering different analysis time-points, may be limited. Our study focused on the year 2021 to also include occupational healthcare data. Migraine affects people of working age and in our study occupational healthcare visits accounted for 17% of total migraine related healthcare contacts. In our study the total one year direct per patient healthcare costs ranged from 2 257 € to 5 626 € depending on patient group. Our study demonstrates that direct per patient healthcare costs of patients suffering from migraine are comparable or even higher compared to other chronic or disabling diseases in Finland.

Previous studies on migraine medication prescription patterns have shown that ~50% of migraine patients have used anti-inflammatory analgesics in Finland during 1998–2006 [22]. In this study, we found that patients belonging to the group with other, or no migraine treatment had relatively high migraine related healthcare resource use compared to patients with two or more triptans. Since triptans have been available in Finland since 1990s', it is possible that this group represents those patients who have discontinued their triptan use before the study period. Similarly, this group may partially represent those patients which use nonsteroidal anti-inflammatory drugs (NSAIDs), or other easily accessible medication not captured by Hilmo and Avohilmo registers. It is plausible that a proportion of patients who have two or more triptan prescriptions may belong to those who have insufficient response to triptans, or patients with more severe form of migraine. The high healthcare resource utilization highlights the importance of finding novel treatment options for these patients.

Our study demonstrated that migraine related HCRU and associated costs increased relatively in the acute treatment group and in the preventive medications group for patients receiving multiple prior medications, with patients with three or more preventive medications having the highest total migraine related HCRU and costs of the studied patient cohorts. Migraine related HCRU and costs were also relatively high for patients with triptan contraindications. As new treatment options, such as CGRPs have been shown to provide favorable health outcomes to patients in clinical studies, they are also associated with higher medicine costs compared to triptans or anti-migraine medications used in preventive care.

Prior studies have suggested CGRPs to be cost-effective in chronic migraine only with highly discounted drug prices [23] or in cohorts that had not responded to multiple prior treatments [24]. In Finland CGRPs are reimbursed with restrictions and confidential price agreements for prophylactic migraine to patients who have an average at least 8 migraine days per month when initiating treatment, and who have tried at least two anti-migraine medications that have not produced an adequate response, are contraindicated, or not tolerated. Furthermore, the eligibility for reimbursement is evaluated periodically based on the reduction of monthly migraine days. To analyze the cost-effectiveness of treatments in each healthcare system it is important to consider the local burden of disease when evaluating which patients could benefit the most from novel treatments, and where potential savings with new treatment options within the healthcare system could be achieved.

The approach chosen to identify migraine was considered most appropriate for population-based registry study, with some limitations. First, disease incidence in the present analysis was based upon the observation of a healthcare encounter for migraine and incident cases were identified based on the appearance of an ICD-10-, ICPC-2- diagnosis code or triptan prescription on a healthcare visit. Therefore, as in most real-world retrospective registry studies, the validity of the results depends upon the accuracy of a physician-assigned diagnosis of migraine and the

resultant diagnostic coding generated by a given encounter. Primary care data coverage is known to be lower than in specialty care [25] some patients do not need medical care for their migraine, and triptan can be prescribed without visit related migraine diagnosis. Unfortunately, despite high prevalence, migraine remains under-diagnosed and inadequately treated [26]. Thus, migraine prevalence in our study is lower than in reality. Furthermore, the severity of migraine could only be determined based on medication treatment as the number of migraine attacks per month is not available from national registers. We have mitigated these limitations by including patients with triptane prescriptions also without migraine diagnosis and according to our knowledge, these limitations do not affect comparisons between patient groups.

The division between patient groups has also some limitations as preventive medication to migraine can be used for other diagnosis and possible medication supply issues might force patients to acquire multiple prescriptions and thus not providing any true estimates of migraine severity. As only patients who had a migraine diagnosis within 2 years were included in the study, some patients are expected to be left out, especially from the Other Migraine Patients group. Also, due to the underdiagnosis of migraine, the groups triptan contraindication and Other Migraine Patients may not represent the "average" migraine patient, but perhaps slightly more severe ones, for whom a migraine diagnosis has been recorded.

A study with migraine patients in Finland demonstrated that the burden of migraine increased by failed treatment lines and was associated with increased comorbidity, with a positive correlation of treatment lines and frequency of comorbidities [16]. The study also showed that in patients with 3 or more prophylactic treatments, a mental disorder such as major depressive disorder or anxiety, was diagnosed in every third person. No data on diagnosis specific medicine purchases are available from the registries included in this study. Thus, in order to reduce the potential confounding of different comorbidities to medicine prescriptions, a medication prescription was tied to a migraine diagnosis related healthcare visit with no more than two weeks prior each prescription. But as our study does not include data on comorbidities within different migraine patient groups, we cannot exclude the possibility that comorbidities such as hypertension, seizures, or mood disorders have not influenced the prescriptions, and consequently the formation of the patient groups. Nevertheless, the mean age and sex distribution of patients treated with acute or prophylactic medications in our study is similar with the previous study of patients with migraine in Finland [16]. Our study also shows similar trends in all-cause, and migraine related healthcare visits presented in that study, demonstrating an increasing burden by each failed treatment and showing the highest burden for patients requiring ≥3 preventive medications.

As all healthcare contacts assessed in our study were physical visits excluding telephone and digital contacts, and as THL registries do not include drug sales data, the total direct burden of migraine associated to society is underestimated. Indirect costs have been found to be greater than direct costs in Europe, accounting for up to 75–91% of the total costs associated with migraine [27–29]. Register studies from Finland have also found that patients refractory to preventive medication had a 3.1-fold increase in absenteeism compared with patients without migraine [16] and that female public sector employees with migraine called in sick an extra 5.4 days per person per year compared to patients without migraine [30]. As the THL registries do not include sick-leaves data, it was not possible to estimate migraine associated indirect costs in this study.

Data for this study was collected on an aggregate level, which means that statistical testing is not possible to the same extent as with patient level data. Due to aggregated level data, HCRU results were calculated from the whole calendar year 2021 despite the fact that some patients in each group did not yet belong to the group at the beginning of the year. As a result, the differences between the groups are likely to appear slightly smaller than they actually are.

## Conclusions

The strength of our study is the national coverage of retrospective registry data covering specialty, primary and occupational healthcare. This study gives novel insights on the distribution of healthcare services utilized by patients in different stages of migraine treatment and what kind of direct costs are associated to these patients in the treatment of migraine in Finland.

Our findings are in line with the recent European Headache Federation (EHF) consensus statement regarding the unmet need in patients who have had inadequate response to two or more triptans [10]. From the healthcare utilization perspective, it is tempting to speculate that early treatment of these patients with novel CGRP medications might help to avoid unnecessary treatment failures, hopefully resulting in decreasing healthcare utilization costs. Indeed, a reduction in service use costs was seen in patients using CGRP preventive medication. Further studies are needed to investigate the potential impact of these treatment options on healthcare resource utilization.

In many healthcare systems resources are currently strained and the need for different healthcare services is most likely increasing due to the aging population. When assessing the patient access and cost-effectiveness of novel treatments for the treatment of migraine within different healthcare systems, a holistic analysis of the current disease burden along with potential gains achieved for patients and healthcare service providers are essential information in guiding decision-making. Nevertheless, the novel treatments targeting CGRP-pathway represent a promising treatment option, especially in those patients previous acute- or preventive migraine medication has failed. In future, studies on the cost-effectiveness of CGRP-pathway targeting medications in the treatment of episodic migraine are warranted.

## Supporting information

**S1 Table. List of triptane contraindications (ICD-10 codes) and migraine preventive medications (ATC-codes).**
(DOCX)

**S2 Table. Total annual all-cause healthcare costs stratified by HCRU components and patient groups.**
(DOCX)

**S3 Table. Total annual migraine healthcare costs stratified by HCRU components and patient groups.**
(DOCX)

**S4 Table. The breakdown of migraine related contacts into different HCRU components.**
(DOCX)

**S5 Table. The breakdown of migraine related costs into different HCRU components.**
(DOCX)

**S1 Data.**
(XLSX)

## Acknowledgments

The authors wish to thank Dr. Marja-Liisa Sumelahti, the chair of Finnish Migraine Current Care Guidelines for defining the triptan contraindication diagnoses.

## Author Contributions

**Conceptualization:** Mikko Kosunen, Jari Rossi, Severi Niskanen, Roope Metsä, Ville Kainu, Mari Lahelma, Outi Isomeri.

**Formal analysis:** Severi Niskanen, Mari Lahelma.

**Funding acquisition:** Jari Rossi.

**Methodology:** Mikko Kosunen, Severi Niskanen, Roope Metsä, Mari Lahelma, Outi Isomeri.

**Project administration:** Mikko Kosunen, Jari Rossi, Roope Metsä, Outi Isomeri.

**Supervision:** Jari Rossi, Outi Isomeri.

**Validation:** Mikko Kosunen, Severi Niskanen, Roope Metsä, Mari Lahelma.

**Visualization:** Mari Lahelma.

**Writing – original draft:** Mikko Kosunen, Jari Rossi, Severi Niskanen, Roope Metsä, Ville Kainu, Mari Lahelma, Outi Isomeri.

**Writing – review & editing:** Mikko Kosunen, Jari Rossi, Severi Niskanen, Roope Metsä, Ville Kainu, Mari Lahelma, Outi Isomeri.

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
