## [Decision Letter · Decision Letter 0]

9 Feb 2024

PONE-D-23-39933Healthcare resource utilization and associated costs among patients with migraine in Finland: A retrospective register-based studyPLOS ONE

Dear Dr. Lahelma,

Thank you for submitting your manuscript to PLOS ONE. After careful consideration, we feel that it has merit but does not fully meet PLOS ONE’s publication criteria as it currently stands. Therefore, we invite you to submit a revised version of the manuscript that addresses the points raised during the review process.

**ACADEMIC EDITOR: **The manuscript has critical concerns to be dealth with. 

We look forward to receiving your revised manuscript.

Kind regards,

Sercan Ergün

Academic Editor

PLOS ONE

Journal Requirements:

Additional Editor Comments (if provided):

The manuscript has critical concerns to be dealth with.

Reviewers' comments:

Reviewer's Responses to Questions

**Comments to the Author**

1. Is the manuscript technically sound, and do the data support the conclusions?

Reviewer #1: Yes

Reviewer #2: Partly

2. Has the statistical analysis been performed appropriately and rigorously? 

Reviewer #1: Yes

Reviewer #2: N/A

3. Have the authors made all data underlying the findings in their manuscript fully available?

Reviewer #1: Yes

Reviewer #2: Yes

4. Is the manuscript presented in an intelligible fashion and written in standard English?

Reviewer #1: Yes

Reviewer #2: Yes

5. Review Comments to the Author

Reviewer #1: The title is very appropriate, and the summary section covers the central aspect of the study. The introduction provides background and information relevant to the research and is very carefully written. The methods are transparent and replicable; all results match the described methods. While the results are not new, they represent progress in this area. The data is plausible. The findings described by the author correlate with the results, and the findings are relevant. The conclusions correlate to the results found. The authors have provided 5 figures and 2 tables; all are clear and legible. I think this article is precious and it would be useful to publish it in your journal.

Reviewer #2: The authors present a retrospective, descriptive analysis of healthcare costs in patients diagnosed with or treated for migraines in a Finnish registry. The authors present two major conclusions: that healthcare costs associated with migraines are significant and likely correlated with disease severity, and that an earlier threshold for novel treatments may be warranted.

The first conclusion is reasonably well-supported by the authors' findings, and is consequential to policy-makers in understanding resource utilization for migraines. One methodological question that could be further clarified relates to patients receiving preventive medications--the authors appear to identify prescriptions for antihypertensives, certain anticonvulsants, amitriptyline, and venlafaxine as preventive treatments for migraines within two weeks of a migraine billing code, although there does not appear to be an analysis of how many of these encounters (or patients) also had billing codes for hypertension, seizures, or mood disorders that could have influenced these prescriptions. Since these medications may well have been selected to address both migraines and a systemic comorbidity, these data points need not necessarily be eliminated, but further insights into potential confounding comorbidities would help the reader better understand the group requiring 3+ preventive medications, which is responsible for disproportionately high migraine-related and all-cause healthcare costs. Compared to the cohorts receiving triptans alone, the groups receiving preventive medications generally appear to have a greater proportion of patients aged 65+, which could again point to confounding from systemic comorbidities.

The supporting evidence for the authors' second assertion regarding earlier use of novel therapies is less clear. The authors rightfully note the "unmet need in patients who have had inadequate response to two or more triptans" and show a modest increase in migraine-related healthcare costs in patients receiving multiple triptans, but omit a discussion regarding the very germane costs of CGRP antagonists as acute therapies for migraines. While these data are limited in episodic migraine, prior studies have suggested cost-effectiveness in chronic migraines only with steeply discounted drug prices (PMID 31302899) or in cohorts that had not responded to multiple prior treatments (PMID 36114468), which would weigh against early use of novel therapies. Moreover, the utility of CGRP antagonists in patients with contraindications to triptans (which had the second-largest healthcare expenditures in the study) remains uncertain given unresolved questions about the effects of CGRP blockade in ischemia. Should the authors wish to include a discussion on the role of novel therapies in episodic migraines, a more nuanced discussion of cost-effectiveness and an additional review of the literature would be appropriate.

In terms of minor revisions:

Line 52 - could be reworded for clarity (e.g., "noted an average of 5.9 doctor's visits per month over the preceding six months")

Line 77 - "attack" may be a more recognizable term to readers than "seizure"

Line 87 - blockade of the renin-angiotensin system appears to apply to antihypertensives rather than anticonvulsants, including in the listed reference

Line 132 - "giant cell arteritis" might be recognizable to a broader audience than "Horton disease"

Line 410 - as noted above, this observation in patients with likely chronic migraines may not apply to those with episodic migraines treated with triptans; additionally, it would be difficult to infer that CGRP antagonists led to a reduction in healthcare costs as the remaining patients on multiple preventive agents do not represent an adequate control group

6. PLOS authors have the option to publish the peer review history of their article (what does this mean?). If published, this will include your full peer review and any attached files.

Reviewer #1: **Yes: **Bilgehan Atılgan ACAR

Reviewer #2: No

---

## [Author Response · Author response to Decision Letter 0]

4 Mar 2024

Please see response to reviewers in the attached 'Response to reviewers.docx'.

---

## [Editor Report · Decision Letter 1]

6 Mar 2024

Healthcare resource utilization and associated costs among patients with migraine in Finland: A retrospective register-based study

PONE-D-23-39933R1

Dear Dr. Isomeri,

We’re pleased to inform you that your manuscript has been judged scientifically suitable for publication and will be formally accepted for publication once it meets all outstanding technical requirements.

Kind regards,

Sercan Ergün

Academic Editor

PLOS ONE

Additional Editor Comments (optional):

Revisions peformed by the authors are well. Revised manuscript is suitable for the publication.
---

## [Editor Report · Acceptance letter]

11 Mar 2024

PONE-D-23-39933R1 

PLOS ONE

Dear Dr. Lahelma, 

I'm pleased to inform you that your manuscript has been deemed suitable for publication in PLOS ONE. Congratulations! Your manuscript is now being handed over to our production team.

Kind regards, 

on behalf of

Dr. Sercan Ergün 

Academic Editor

PLOS ONE